

# Effects of bortezomib on intracellular antioxidant and apoptosis in HepG2cells

Grażyna Świderska-Kołacz[1], Magdalena Madej[1], Szymon Zmorzynski[2], Wojciech Styk[3], Iwona Surowiec[1], Bożena Witek[1], Anna Wojciechowska[4], Joanna Czerwik-Marcinkowska[1] and Anna Nowakowska[5]

[1] Institute of Biology, Jan Kochanowski University, Kielce, Poland
[2] Institute of Human Sciences, Academy of Zamość, Zamość, Poland
[3] Academic Laboratory of Psychological Tests, Medical University of Lublin, Lublin, Poland
[4] Department of Geobotany and Landscape Planning, Nicolaus Copernicus University of Torun, Toruń, Poland
[5] Department of Animal Physiology and Neurobiology, Nicolaus Copernicus University of Torun, Toruń, Poland

## ABSTRACT

Bortezomib, as a proteasome inhibitor, is used in clinical trials related to solid cancers. However, its use is not always associated with a good response to treatment. Taking into account the above, we decided to analyze the effect of the time-dependency (24 *vs.* 48 h) and the dose-dependency of bortezomib (2, 4, 8 and 16 nM) on apoptosis and activities of antioxidant enzymes such as catalase (CAT), superoxide dismutase (SOD), glutathione reductase (GR), glutathione peroxidase (GPx) and glutathione transferase (GST), as well as concentrations of reduced glutathione (GSH) and malondialdehyde (MDA) in hepatoblastoma cell line (HepG2) cells. We have shown that increasing concentrations of bortezomib caused (I) a gradual decrease in the levels of GSH; (II) changes in MDA concentrations and antioxidant enzymes activities; (III) increase in apoptosis levels in HepG2 cells. We did not find significant association between antioxidant parameters and number of apoptotic cells. Our study showed that the analyzed parameters (such as: CAT, SOD, GR, GPx, GST, GSH, MDA) changed after bortezomib treatment. It is important to search for new anti-cancer therapies based on next-generation proteasome inhibitors. It is possible that the use of proteins associated with oxidative stress will help enhance the action of these inhibitors and will provide a better treatment effect.

# INTRODUCTION

Proteasome inhibitors are a promising class of compounds that reduce the ability of cancer cells to cope with the increase in by-products obtained from protein synthesis, such as the accumulation of misfolded proteins, leading to cell death (*Manasanch & Orlowski, 2017*). The bortezomib, a potent proteasome inhibitor, is a boronic acid dipeptide that induces cancer cell death (*Park et al., 2018*). It is an effective drug for both newly diagnosed or relapsed multiple myeloma patients and is widely used in the treatment of various kinds of lymphomas, solid cancers and other diseases. In light of their success

Corresponding author
Joanna Czerwik-Marcinkowska,
joanna.czerwik-marcinkowska@ujk.edu.pl

in hematologic malignancies, proteasome inhibitors have been evaluated in various solid tumours, including hepatocellular cancer (HCC) (*Kim & Viatour, 2020*; *Roeten, Cloos & Jansen, 2017*). There are several studies showing the mechanisms of bortezomib action (called Velcade or PS341) in the HepG2 cell line (hepatoblastoma cell line), including the application of HepG2 in an animal model (*Yang et al., 2016*). Bortezomib affects the function and signalling of HepG2 cells (*Calvaruso et al., 2006*; *Baiz et al., 2009*; *Liao et al., 2023*). In preclinical studies using HCC cell lines, proteasome inhibitors induced various antitumor responses, such as cell cycle arrest, apoptosis, induction of endoplasmic reticulum stress, repression of NF$\kappa$B signalling, and inhibition of epithelial-mesenchymal transition (*Augello et al., 2018*; *Huang et al., 2019*). Bortezomib induces apoptosis in HepG2 cells by both extrinsic and intrinsic pathways (*Lauricella et al., 2006*). Moreover this drug can cause oxidative stress development *via* cytochrome in mitochondrial fraction. However, the exact effect of bortezomib on glutathione enzymes and other antioxidant parameters in HepG2 cells was not analyzed. Bortezomib affects the redox homeostasis in multiple myeloma cells decreasing the levels of intracellular glutathione (*Caillot et al., 2021*; *Nerini-Molteni et al., 2008*; *Abou-Ghali & Stiban, 2015*). Reduced glutathione (GSH) is one of the most important endogenous antioxidants. It is found in the mitochondria, nucleus, endoplasmic reticulum and the cytoplasm. Cellular GSH protects DNA, proteins and fats against the harmful effects of free oxygen radicals. In addition to its important role in maintaining redox homeostasis, it regulates metabolic processes and affects apoptosis (*Greenberg, 1996*; *Paduch, Klatka & Klatka, 2015*). Glutathione interacts with enzymes such as: glutathione S-transferases (GSTs), glutathione reductase (GR) and glutathione peroxidase (GPx). GSTs reduce or eliminate the toxicity of xenobiotics and exogenous toxins. GPx promotes the reduction of superoxide hydrogen to water using GSH as a reducing compound (*Flohé, Toppo & Orian, 2022*). As a result of this reaction oxidized glutathione (GSSG) is produced. GR catalyzes GSSG reduction using NADPH as a reducing agent (*Caillot et al., 2020*). The first line of cellular antioxidant defence is formed not only by GPx, but also by superoxide dismutase (SOD) and catalase (CAT) (*Caillot et al., 2020*). Superoxide dismutases are the only enzymes that can eliminate superoxide radicals by catalysing their dismutation into hydrogen peroxide and oxygen (*Greenberg, 1996*). CAT catalyses hydrogen peroxide transformation into water and oxygen (*Caillot et al., 2020*). The level of oxidative stress is associated with high cytotoxic potential of malondialdehyde (MDA), which acts as a carcinogen (*Gubaljevic et al., 2018*). MDA concentration is a cytotoxic substance resulting from lipid peroxidation, oxidising glutathione, cysteine, and SH group proteins. In addition, it inhibits the action of enzymes and membrane proteins and is one of the markers of oxidative stress (*Khoubnasabjafari, Ansarin & Jouyban, 2015*).

In recent years, molecular targeted anticancer drugs with fewer side effects than classical anticancer drugs have applied for cancer therapy (*Zhang et al., 2020*).

HCC is the seventh most common cancer and the third most common cause of cancer-related mortality worldwide (*Hentze et al., 2002*; *Circu & Aw, 2008*). The majority of patients with HCC suffer from liver dysfunction and cannot be treated with intensive chemotherapy (*Bray et al., 2001*); thus, there is a need to develop novel therapies for HCC. The lack of effective treatment options, especially for patients with advanced and

unresectable HCC is a significant clinical challenge. The lack of effective treatment options for advanced HCC requires ongoing research into new treatments and new therapeutic targets (*Lim et al., 2022*). HepG2 cells are also used as a model system for studies concern liver metabolism, xenobiotics toxicity, and chemotherapeutic drugs effects including apoptosis induction (*Evan & Vousden, 2001*). Additionally, this line is often used in research due to the very low proliferative potential of normal hepatocytes. Apoptosis can be generated by a number of different factors of external origin (extrinsic apoptosis pathway), or a condition inside the cell (intrinsic apoptosis pathway). The intrinsic apoptotic pathway is activated by various intracellular stimuli, such as oxidative stress (*Pfeffer & Singh, 2018*; *Modanloo & Shokrzadeh, 2019*).

Taking into account above, we have decided to analyse the relationship between the activity/concentration of selected oxidative stress-related substances (glutathione and its enzymes, as well as catalase, superoxide dismutase and malondialdehyde). HepG2 cells were exposed to various bortezomib doses and incubation times. The hypothesis of the research is as follows: the activity of antioxidant enzymes and concentrations of reduced glutathione, as well as malondialdehyde depends on bortezomib doses and incubation time with this drug. The higher concentration of bortezomib and the longer incubation time should result in greater dynamics of changes in activity of the antioxidant enzymes. Furthermore, the correlation between the level of apoptosis and antioxidant markers was analyzed. According to our knowledge aforementioned parameters were not analyzed to such an extent in HepG2 cells in the context of bortezomib concentrations. The work aimed at evaluating the impact of bortezomib on redox dynamics of human HCC cells, what can be used in the future to develop new treatment strategies.

# MATERIALS AND METHODS

## Cell cultures and bortezomib treatment

The study was conducted on the human hepatoma cell line (HepG2) (from the Institute of Biology at Jan Kochanowski University in Kielce). The cells were plated $2.0 \times 10^5 - 3.0 \times 10^5$ per dish in one mL of complete growth medium, according to manufacturer's recommendations (ATCC, Manassas, VA, USA). Bortezomib was added to the culture medium at concentrations of: 2 nM, 4 nM, 8 nM and 16 nM. Bortezomib (LC Laboratories, Woburn, MA, USA) 10 mg (free base, B-1408) was dissolved in DMSO (with final concentration 200 mg/ml, as recommended by the manufacturer) and stored at $-80\,°C$. The final DMSO (Merck, Burlington, MA, USA) concentration in culture medium was less than 0.1%. As a control, cell cultures without bortezomib (with 0.1% DMSO) were used. The cultures were incubated in an atmosphere of 5% $CO_2$, at 37 °C for 24 h or 48 h. Passaging procedures were repeated until 12 dishes were obtained for each concentration and control. The cell pellet was suspended in a medium consisting of: 2.4 mL of HepG2 (William's Medium) A549 (F12-HAM), 0.9 mL of FCS and 0.4 mL of DMSO.

## Apoptosis/necrosis determination

In order to determine the apoptosis level in HepG2 cells, the commercial FITC Annexin V Apoptosis Detection Kit I from BD Pharmingen Sigma-Aldrich (Darmstadt, Germany)

was used, following the manufacturer's instructions. Trypsin (0.25%) was added to collect the cells of all groups, and the cell density was adjusted to $1 \times 10^6$ cells/ml. Annexin V-FITC and PI were added, respectively, for dyeing before analysis with flow cytometry. The effect of bortezomib variation on cell apoptosis by annexin V-FITC/PI staining and flow cytometry was evaluated. The cells are double stained with annexin V/PI and three different populations of cells were observed. The cells which were not stained with both dyes were alive (viable) and resided in region Q3. The cells the stage of early apoptosis that were stained with only annexin V, resided in region Q4. The cells that were stained with both reagents were in late apoptosis and scattered in region Q2. The cells in region Q1 were undergoing necrosis. The cells that stain with both reagents are in late apoptosis. But the cells that stain with only annexin V are in the stage of early apoptosis. Early apoptotic cells are annexin V-positive and PI-negative (annexin V-FITC+/PI−), whereas late (end-stage) apoptotic cells are annexin V/PI-double-positive (annexin V-FITC+/PI+). However, to verify the stages of apoptosis, time-course analyses and caspase assays are necessary. Necrosis is a nonapoptotic, accidental cell death. It is a term used to designate the presence of dead tissues or cells and is the sum of changes that have occurred in cells after they have died, regardless of the prelethal processes. Necrosis, therefore, refers to morphological stigmata seen after a cell has already died and reached equilibrium with its surroundings. Thus, in the absence of phagocytosis, apoptotic bodies may lose their integrity and proceed to secondary or apoptotic necrosis. Measurement was performed with a Becton Dickinson LSR II flow cytometer using the CellQuest Pro computer system from Becton Dickinson, determining the percentage of cells during early and late apoptosis and necrosis.

## Determination of cell viability (MTT test)

After 24-hour or 48-hour incubation with bortezomib (2, 4, 8 and 16 nM) the culture medium was removed and the cells were resuspend in tetrazolium salt 3-(4,5-dimethylthiazol-2-yl)-2,5-diphenyltetrazolium bromide or MTT in concentration (0.5 mg/ml PBS). The incubation with MTT was for 3 h at 37 °C. The formed formazan crystals were dissolved in dimethyl sulfoxide (DMSO) and the absorbance was read at 570 nm using a TECAN ELISA microplate reader. The cell viability was then calculated as a percent of the control. It was done by dividing the absorbance of the cultures treated with bortezomib by the absorbance of the control cultures. The resulting number was multiplied by 100 to give a percentage.

## Analysis of enzymes activities
### Superoxide dismutase

SOD activity measurements were performed using the adrenaline method according to *Misra & Fridovich (1972)*. The absorbance changes were measured at a wavelength of 480 nm against black samples containing 2,000 µL of 0.05M carbonate buffer (pH 10.2) and 1,000 µL of 0.3 mM EDTA. The control sample contained: 0.05 M carbonate buffer pH 10.2 (1,900 µl), 0.3 mM EDTA (1,500 µl) and 9 mM adrenaline (200 µl). The samples used for assessing enzymes activity contain 1,800 µl of 0.05 M carbonate buffer (pH 10.2), 1,500 µl of 0.3 mM EDTA, 100 µl of the enzyme extract and 200 µl of 9 mM adrenaline, which

was added immediately before the measurement. The activity of SOD was determined at one-minute time intervals based on changes in absorbance in the sample containing the enzyme, in relation to analogous time changes in absorption in the control sample (measurement for 3 min). SOD activity is expressed in U/mg protein/min.

### Catalase

The measurement of catalase activity was assessed on the basis of the rate of decomposition of 54 mM $H_2O_2$ hydrogen peroxide in 50 mM phosphate buffer (pH 7.0) and 20 µL of the assayed enzyme extract in a total volume of three mL according to *Bartosz (2006)*. The absorbance was measured at a wavelength of $\lambda = 240$ nm over a period of 3 min. CAT activity was expressed in U/mg protein/min. The unit of catalase is defined as the reduction of 1 µmol/L of the peroxide per minute. For details of the methods used to assess activity of CAT see *Nowakowska, Caputa & Rogalska (2011)*.

### Glutathione peroxidase

Activity of glutathione peroxidase was measured with a Glutathione Peroxidase Cellular Activity Assay Kit (Sigma-Aldrich, Darmstadt, Germany). Extinction was measured spectrophotometrically in a kinetic program at a wavelength of $\lambda = 340$ nm every 15 s for 1 min. GPx activity was expressed in U/mg protein/min.

### Glutathione reductase

Glutathione reductase activity was measured with Glutathione Reductase Assay Kit (Sigma-Aldrich, Darmstadt, Germany). The activity was measured by the increase in absorbance caused by the reduction of DTNB [5.5″-dithiobis(2-nitrobenoic acid)] at 412 nm. GR activity was expressed in U/mg protein/min.

### Glutathione transferase

Glutathione transferase was measured with the Glutathione S-Transferase (GST) Assay Kit (Sigma-Aldrich, Darmstadt, Germany). GST catalyses the conjugation of L-glutathione to the 1-chloro-2,4-dinitrobenzene (CDNB) through the thiol group of the glutathione at 340 nm. The rate of increase in the absorption is directly proportional to the GST activity in the sample. GST activity was expressed in U/mg protein/min.

## Determination of reduced glutathione concentration

The concentration of reduced glutathione was determined using the Sigma Aldrich Glutathione Assay Kit (Darmstadt, Germany). Fluorometric test (CS0260 1 KT) was performed using the TECAN ELISA microplate reader at the wavelength $\lambda = 412$ nm. Total protein concentration was determined according to the method by *Lowry et al. (1951)*. The GSH concentration was expressed in µM/mg protein.

### Protein content

Protein content was determined using the method by *Lowry et al. (1951)* using bovine serum albumina (Sigma Chemical, Steinheim, Germany) as a standard. Absorbance was read at a wavelength of $\lambda = 750$ nm against a reagent blank (500 µl and 2.5 ml of solution).

## Determination of malondialdehyde concentration

The MDA concentration was measured spectrophotometrically (ELISA, TECAN) using the thiobarbituric (TBA) acid assay (Lipid Peroxidation MDA Assay Kit). An excitation source with a wavelength of $\lambda = 532$ nm was used. Results were expressed in nmol/μl. For each sample, two independent replicates of the experiment were performed in relation to the blank sample and the properly prepared MDA standard. The MDA concentration was expressed in nmol/μl. All assays for oxidative stress factors were performed on Specol Genesys 10SUV-V spectrophotometer but protein contents were run on EVOLUTION 300B spectrophotometer.

## Statistical analysis

All results concerning oxidant/antioxidant status of HepG2 cells are presented as mean of four experiment repetitions. The values obtained in the individual experimental groups were compared to the control group, which consisted of cells not treated with bortezomib. The significance of differences between indices of oxidative stress (SOD, CAT, GPx, GSH, MDA and protein control) was determined using the non-parametric Kruskal–Wallis test. The non-parametric test was used due to small numbers in groups. These analyzes and the graphical representation of dependencies were performed using Statistica 9.0 software (StatSoft, Inc., Tulsa, OK, USA). Detailed comparisons were made using Dunn's *post hoc* test. To determine the association between bortezomib treatment and cell survival, a Chi$^2$ test was applied. These tests were performed in PAST 4.16c (*Hammer, Harper & Ryan, 2001*). To determine whether the differences in the MTT test results for different bortezomib concentrations were statistically significant, one-way ANOVA and Tuckey's test as *post-hoc* were performed (*McHugh, 2011*). Normality tests (Shapiro-Wilk tests) were also performed and parametric tests were performed based on their results. Results of MTT test were presented as graphs with standard error and standard deviation. PAST 4.16c (*Hammer, Harper & Ryan, 2001*) programs were used. Data on the rate of apoptosis in relation to time and bortezomib concentration were used to perform a direct ordination analysis (CCA, Canonical Correspondence Analysis). To determine which of the studied variables were statistically significant for the rate of apoptosis, a Monte Carlo permutation test was performed during the CCA. The result of CCA is an ordination diagram in which the percentage of cells in each experimental variant is marked with geometric symbols and time and bortezomib concentration (and control) are marked with vectors. The analysis was performed in the Canoco 5.0 program (*Ter Braak & Šmilauer, 2012*).

# RESULTS

## Effect of bortezomib on apoptosis in HepG2 cells

The examples of flow cytometry analysis after bortezomib treatment are shown in Fig. 1. The results obtained from all experimental groups are shown in Fig. 2.

The effect of bortezomib variation on cell apoptosis by annexin V-FITC/PI staining and flow cytometry was evaluated. The cells are double stained with annexin V/PI and three different populations of cells were observed. The cells which were stained with both dyes were alive (viable) and resided in region Q3. The cells the stage of early apoptosis that were

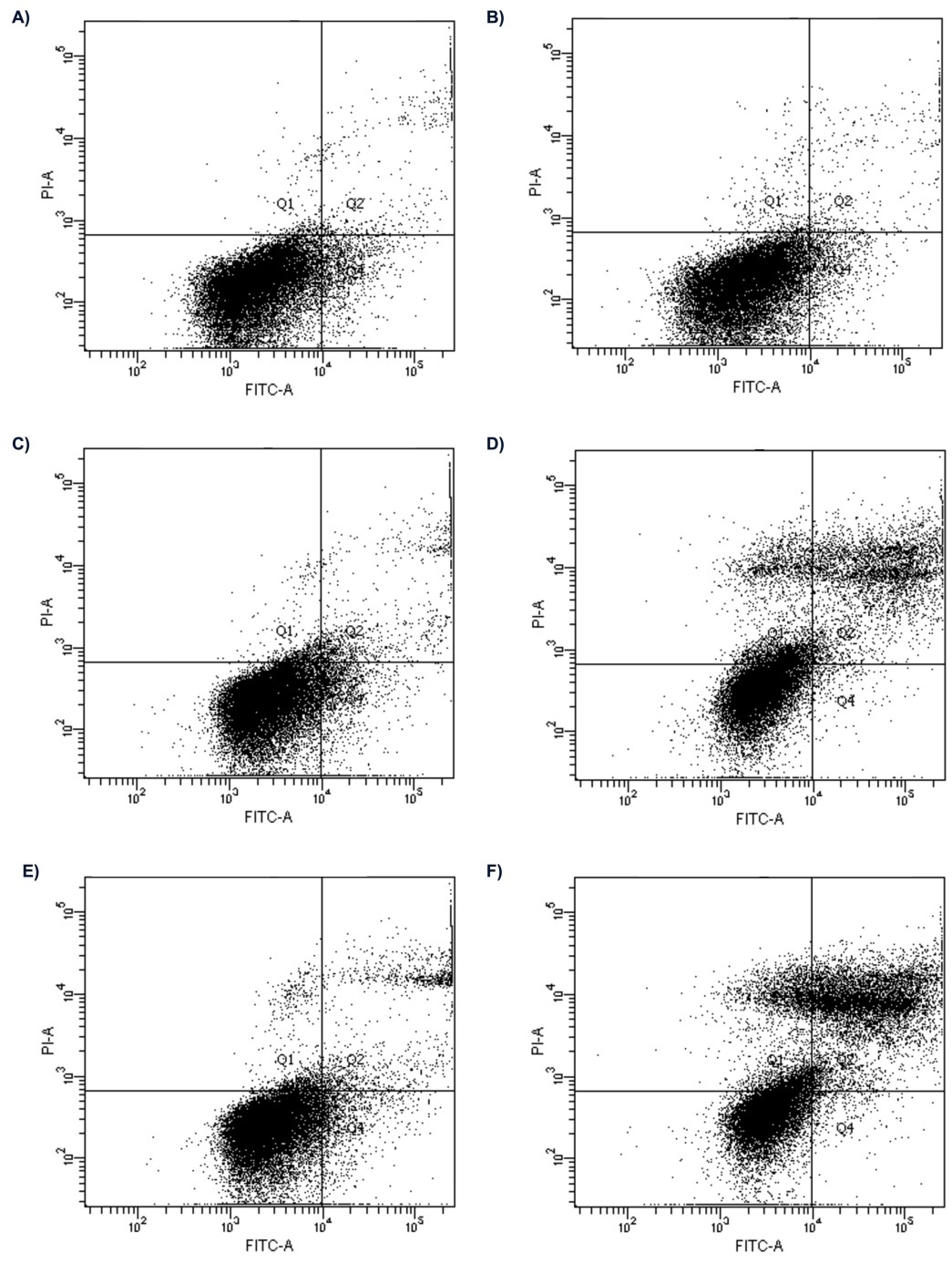

**Figure 1** **Flow cytometry analysis of apoptosis in HepG2 cells treated with 2 nM and 16 nM of bortezomib.** Control for (A) 24 h; (B) 48 h; the cells treated with 2 nM of bortezomib for (C) 24 h, (D) 48 h; and 16 nM of bortezomib (C) for 24 h and, (D) 48 h.

stained with only annexin V, resided in region Q4. The cells that were stained with both reagents were in late apoptosis and scattered in region Q2. The cells in region Q1 were undergoing necrosis. Flow cytometry analysis showed apoptosis induced by bortezomib

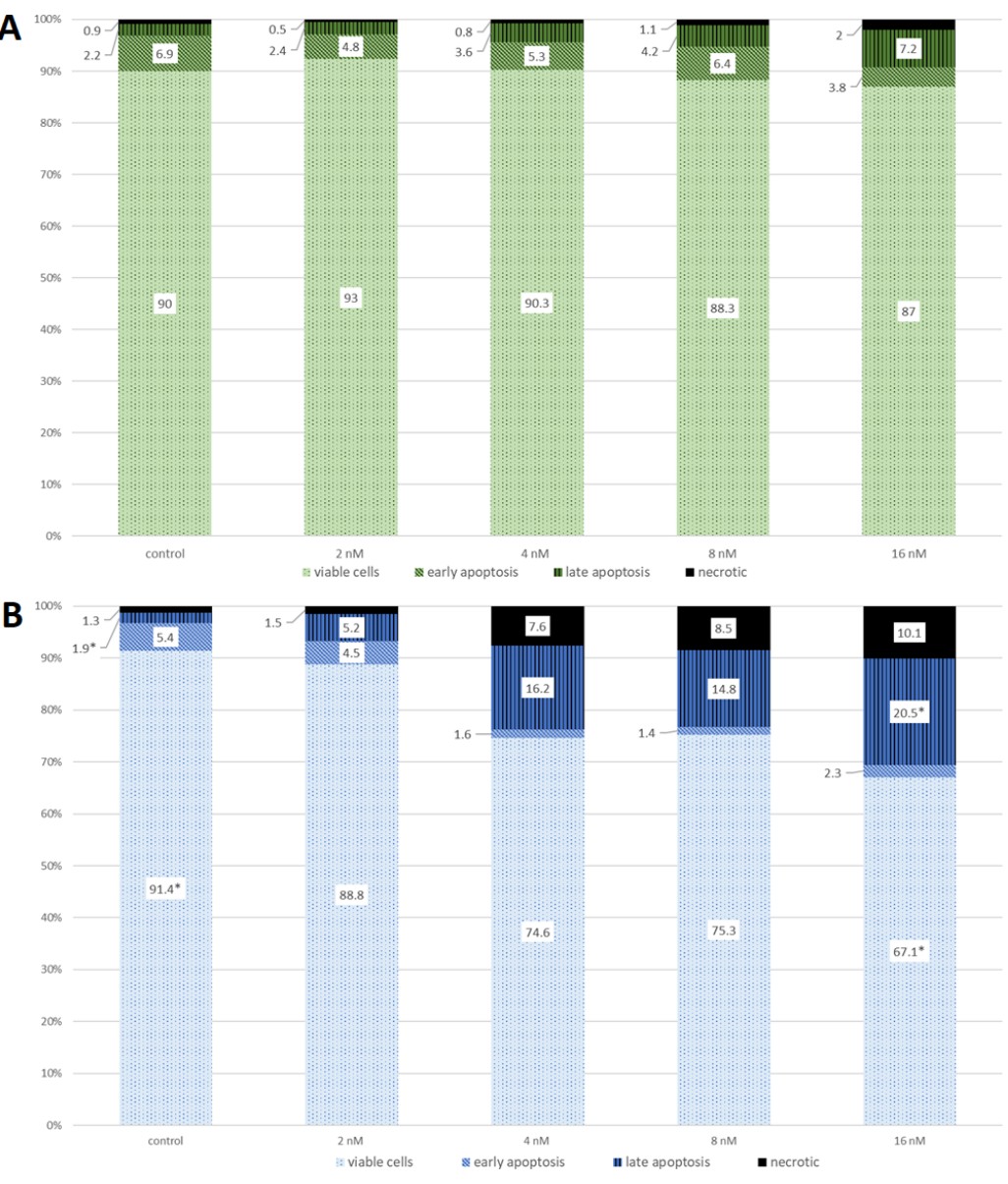

**Figure 2** **The cell viability obtained by MTT after 24 (green) and 48 (blue) h of treatment with borte-zomib.** (A) The cell viability obtained by MTT after 24 h of treatment with bortezomib. (B) The cell via-bility obtained by MTT after 48 h of treatment with bortezomib.

in HepG2 cells in a time-dependent and dose-dependent manner (Figs. 1 and 2). HepG2 cells were incubated with bortezomib for 24 or 48 h and cells in late and early apoptosis were analyzed. After 24 h the lowest number of cells in early apoptosis compared to the control group was observed at bortezomib dose of 16 nM (Figs. 1A and 1C, 2A). Stronger effect of bortezomib on early apoptosis induction was recorded after 48 h at doses of 4 nM ($p < 0.001$), 8 nM ($p < 0.001$) and 16 nM ($p < 0.001$) (Fig. 2B). After 24 h higher number of cells in late apoptosis was identified at bortezomib concentrations of 4 nM, 8 nM and

16 nM (Fig. 2A). After 48-hour incubation with bortezomib higher number of cells in late apoptosis was observed at all bortezomib concentrations. After 24-hour treatment the concentrations of bortezomib (16 nM) increased the number of necrotic cells (2% *vs.* 0.9% in control group). After 48-hour incubation similar effects (compared to the control group) were observed at bortezomib concentrations of 4 nM (7.6% *vs.* 1.3%), 8 nM (8.5% *vs.* 1.3%) and 16 nM (10.1% *vs.* 1.3%).

The Chi$^2$ test indicated no significant results between bortezomib doses and cell survival after 24 h. However, after 48 h, this test showed a significant association between the number of viable cells in the control (most) and a significant association between the number of cells with late apotosis in the control (least). The same parameters were significantly associated with the 16 nM concentration of bortezomib, with viable cells being the least and those with late apoptosis the most.

## Changes in antioxidant status of HepG2 cell

Increasing concentrations of bortezomib caused a gradual decrease in the level of GSH. Among all experimental groups the lowest level of GSH (Fig. S1) was recorded after 24- and 48-hours exposure to bortezomib at a concentration of 16 nM and the highest level was found in both control groups (incubated 24- and 48-hours in an atmosphere of 5% $CO_2$, at 37 °C). Further analysis with Kruskal–Wallis and Dunn's as a *post hoc* tests revealed that statistically significant drops in concentrations of intracellular GSH (in comparison to control) were recorded after 48-hours exposure to bortezomib at a concentration of 4 nM, as well as after 24- and 48-hours exposure to bortezomib at concentrations of 8 nM and 16 nM. Neither 24- nor 48-hours incubation with 2 nM bortezomib evoked any effect on GSH concentration. Similarly, 24-hours incubation with 4 nM bortezomib did not induced changes in GSH concentration. Exposure to bortezomib at concentrations of 8 nM and 16 nM, regardless of the incubation time, causes the same effect on glutathione concentration.

The effect of bortezomib on GPx activity was inconsistent (Fig. S2). The incubation time affected GPx activity in the cell line not treated with bortezomib as well as in that treated with bortezomib at a concentration of 8 nM. In both cases there were statistically significant decreases in GPx activity after the prolonged exposure. A significant decrease in GPx activity in relation to that in the control group was observed only after 24-hours incubation with bortezomib at concentrations of 8 nM and 16 nM. The 48-hours incubation caused a decrease in activity of GPx but only at bortezomib concentrations of 4 nM, 8 nM and 16 nM.

In control group as well as in all experimental groups incubation time affected GST activity (Fig. S3). The 48-hours incubation induced significant decrease in GST activities comparing to those recorded after 24-hours. Bortezomib at concentrations of 8 nM and 16 nM reduced significantly the activity of GST, compared to that in control group, as well as in groups with bortezomib at concentrations of 2 nM and 4 nM after 24-hours incubation. However, after the prolonged exposure, bortezomib at concentrations of four, eight and 16 nM reduced GST activity compared to that recorded at the concentration of two nM.

The highest activity of GR was recorded in HepG2 cells treated with bortezomib at concentrations of two nM and four nM (Fig. S4). The GR activities after 24- and 48-hours incubation with bortezomib at concentrations of two nM and four nM changed statistically significantly compared to that recorded in the control groups, respectively. However, there were no statistically significant differences between 24- and 48-hours incubation at these concentration.

Time of incubation affected CAT activity in control group as well as in cells that treated with bortezomib at a concentration of two nM (Fig. S5). After the prolonged incubation, the highest CAT activities were recorded in both groups. Activity of CAT after 48-hours incubation with bortezomib at a concentration of two nM was significantly higher than those recorded after 48-hours incubation with bortezomib at a concentration of 16 nM and in control group. The lowest activities of CAT were recorded 24-hours incubation with bortezomib at concentrations of eight nM and 16nM but the values were not statistically significant.

The 48-hours incubation affected SOD activity in control group as well as in that treated with bortezomib at a concentration of two nM but the increase was statistically significant only in the control group (Fig. S6). The lowest activities of SOD were recorded at bortezomib at concentrations of eight nM and 16 nM but the decreases were significant statistically after 48-hours incubation. Time of incubation had no effect on the activity of SOD in any of the experimental groups.

The highest levels of MDA concentration were recorded in groups treated with bortezomib at concentrations of two nM and four nM after 48-hours incubation (Fig. S7). In both cases the concentration of MDA was significantly higher than that in the control group. In neither group, a 24-hours incubation resulted in an increase in MDA concentration.

The results of one-way ANOVA for the MTT test, for 24 h and 48 h are analogous. They indicate that the percentage of live cells is the highest at a concentration of two nM and significantly different from the lowest value at a concentration of eight nM (Figs. 3A and 3B). CCA analysis indicated that significantly more live cells were associated with bortezomib concentration of two nM and shorter exposure time. There was also a significant association with increasing number of dead cells after 48 h of exposure at the highest bortezomib concentration (Fig. 4).

## DISCUSSION

In this study, we have explored the association of bortezomib concentrations with the activity/concentration of selected oxidative stress factors (glutathione and its enzymes, as well as catalase, superoxide dismutase and malondialdehyde) in HepG2 cell line. To the best of our knowledge, this is the first study elucidating the correlation between activity/concentration of the factors and the drug-induced apoptosis of HepG2 cells.

Many studies have showed that bortezomib can reversibly suppress the proteasome pathway by binding with the 20S proteasome complex directly and blocking its enzymatic activity (Xing et al., 2017). Moreover, bortezomib concentration (two nM) was effective

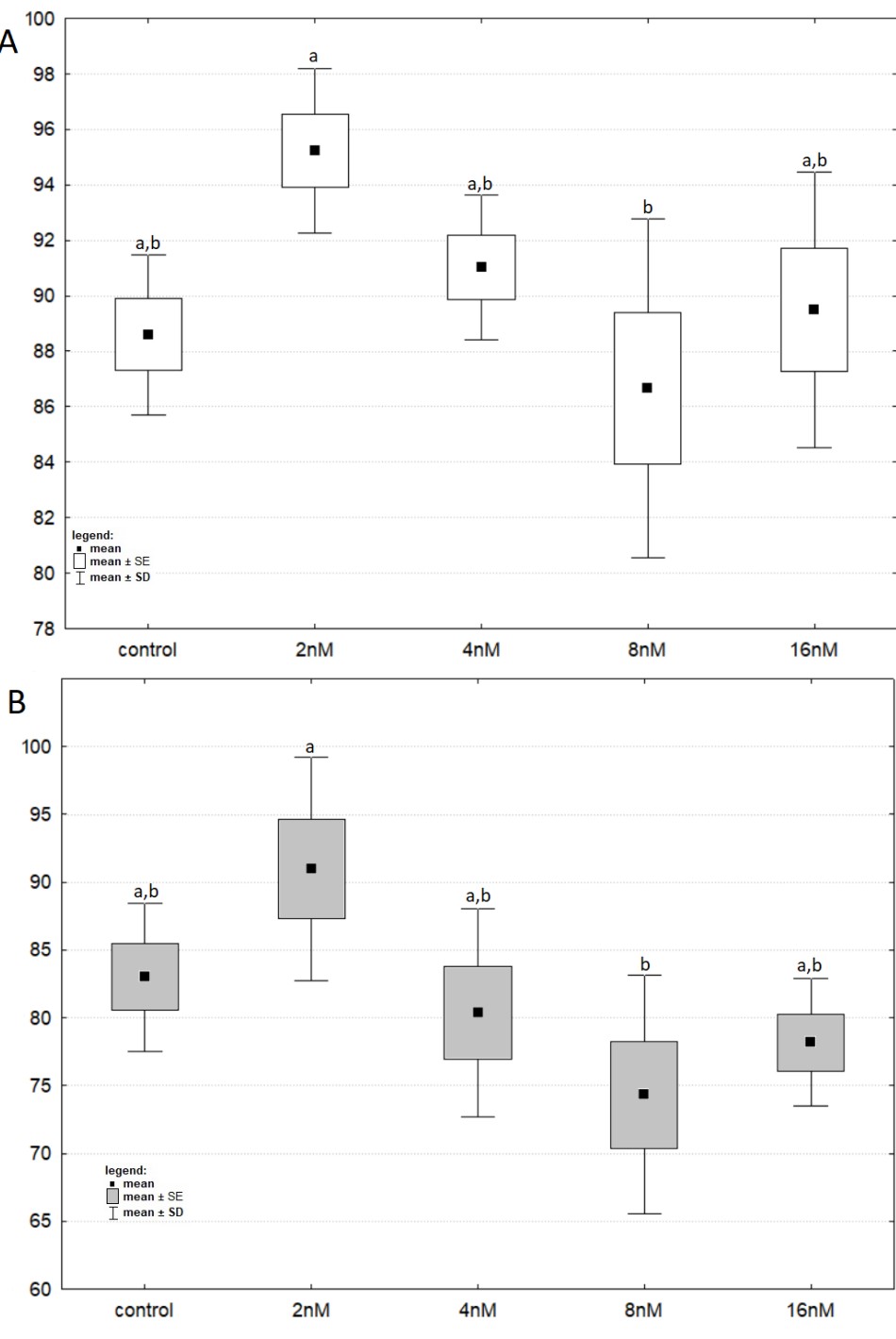

**Figure 3   The cell viability obtained by MTT after 24 and 48 h of treatment with bortezomib.** (A) The cell viability obtained by MTT after 24 h of treatment with bortezomib. (B) The cell viability obtained by MTT after 48 h of treatment with bortezomib.

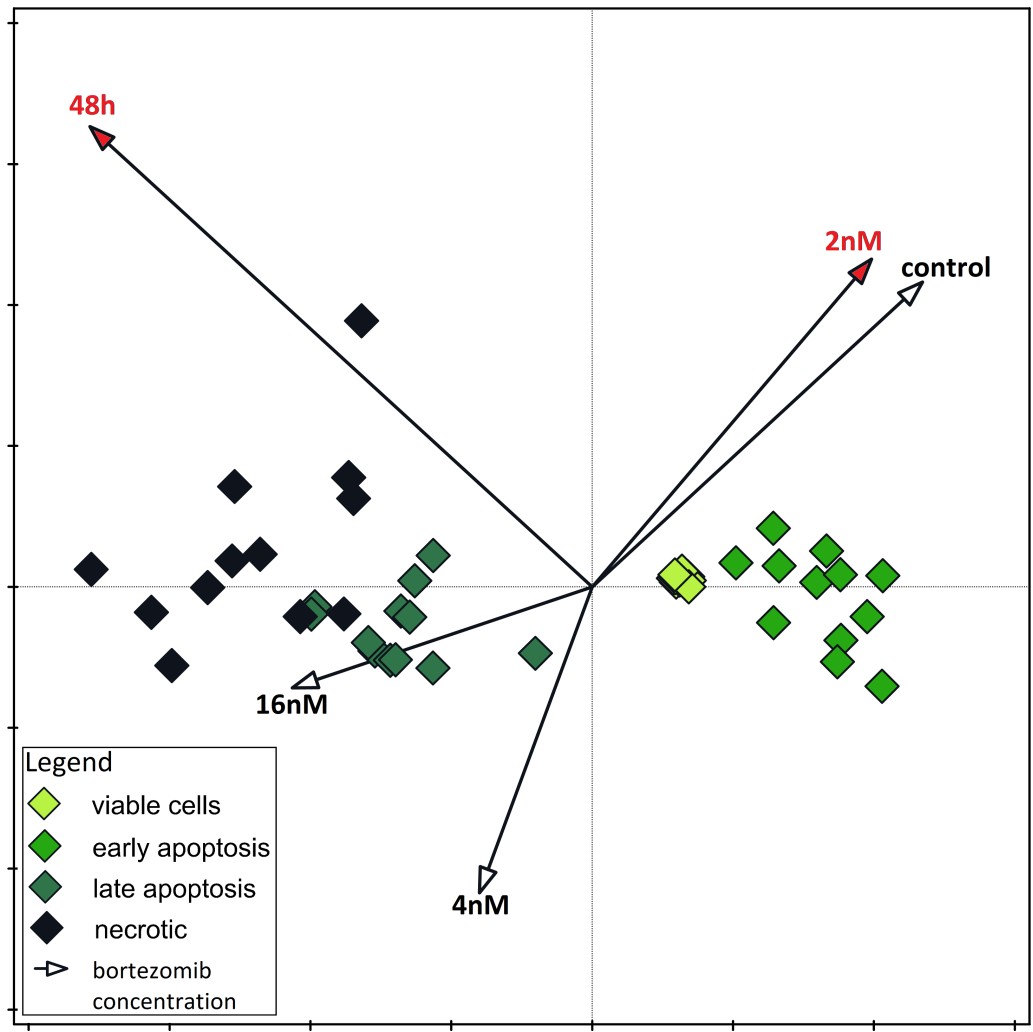

**Figure 4 Canonical correspondence analysis.** Canonical correspondence analysis of data on the dependence of apoptosis rate on time and bortezomib concentration. Variables significant for the variability in the data set are marked in red (Monte Carlo Permutation Test, $p < 0.05$).

in most cell-based models starting from research on PS-341, before this inhibitor was accepted for use in human (*Reece et al., 2011*). *Zmorzyński et al. (2024)* provided an effect of bortezomib on reduced glutathione (GSH) and the activity of glutathione enzymes in multiple myeloma cells. Bortezomib increased the number of apoptotic cells and decreased the activities of S-glutathione transferase (GST) and glutathione peroxidase (GPx).

The induction of the apoptosis process is one of the main goals of anticancer therapies. This a challenge is due to efficacious mechanisms of tumor cell resistance to the programmed death. Recent studies have shown that oxidative stress induces cell apoptosis in physiological, as well as pathological states through both mitochondria-dependent and mitochondria-independent pathways (*Radi et al., 2014*; *Kane et al., 2006*; *Hanikoglu et al., 2020*; *Fan et al., 2021*).

The accumulation of unfolded proteins within the cell can increase the rate of oxidative stress and induce apoptosis (*Sharma et al., 2006*; *Akhigbe et al., 2022*). The proteasome system is involved in degradation of unfolded proteins (*Starheim et al., 2016*). However, the activities of proteasomes in cancer cells are higher than in normal cells. Furthermore, a number of proteasome substrates, which are involved in the cell cycle or apoptosis, have been identified (*Obeng, 2006*). Preclinical and clinical trials in both hematological malignancies and solid tumors have demonstrated that bortezomib is a relatively well-tolerated drug and can act in combination with traditional chemotherapeutic drugs (*Holmström & Finkel, 2014*). Bortezomib promotes the cell death *via* overproduction of reactive oxygen species (ROS) (*Wu, Yang & Saitsu, 2016*; *El-Kenawi & Ruffell, 2017*). *Wu, Yang & Saitsu (2016)* found that resistance to bortezomib can be achieved in the HCC cell line. Moreover, the high basal levels of proteasome activities in bortezomib-resistant HCC cells are due to increased expression of proteasome subunits and bortezomib-resistant HCC cells acquire resistance to apoptosis by losing the ability to stabilize and accumulate pro-apoptotic proteins. The production of ROS increases the number of mutations in genetic material and causes malignant transformation or cell death (*Aykin-Burns et al., 2009*; *Canli et al., 2017*). Carcinogenesis leads to development of cancer cells, which frequently show altered oxidative metabolism inducing the initiation of bortezomib-induced apoptosis (*Perez-Galan, 2006*; *Cong, 2019*). The changes in the activity and concentration of intracellular antioxidants are associated with higher susceptibility to bortezomib-induced apoptosis (*Reece, 2011*; *Zmorzyński et al., 2019*). As a part of our study, we examined the effect of bortezomib doses on apoptosis in HepG2 cells. The applied experimental model was designed to check time- and dose-dependent effect of bortezomib on cell viability. Our investigation confirmed that bortezomib reduced the number of viable cells. *Lauricella et al. (2006)* have shown that bortezomib induced apoptosis in HepG2 cells is due to stimulating both the extrinsic and intrinsic apoptotic pathways. However, they examined the apoptotic effect induced by proteasome inhibitor (Cbz-leu-leu-leucinal) (MG132) in human hepatoma HepG2 cells at bortezomib concentration of 50 nM, which induced apoptotic effects after a lag phase of 16–24 h. In our study bortezomib increased the number of cells in late apoptosis after 48-hours incubation (Fig. 1). It is consistent with results obtained by other researchers (*Kane et al., 2006*; *Trachootham, Alexandre & Huang, 2009*). The redox-directed therapies inhibiting activity of antioxidant enzymes, as well as decrease in the concentration of antioxidant compounds such as reduced glutathione have been offered to induce cytotoxicity in cancer cells (*Schafer & Buettner, 2001*; *Salem et al., 2015*). In our investigation bortezomib decreased GSH concentration after 24 and 48 h (Fig. S1). This suggests a possible approach to using this drug in clinical trials for liver cancers. *Nerini-Molteni et al. (2008)* analyzed the relationships between redox homeostasis and bortezomib treatment in MM cells. They have shown that decreasing intracellular glutathione strongly enhances bortezomib toxicity, whilst antioxidants protect MM cells from bortezomib-mediated cell death, According to *Hanikoglu et al. (2020)*, the decrease in GSH impairs the antioxidant system and leads to an increase in ROS production, which accelerates mitochondrial damage and induces apoptosis. Bortezomib-resistant cells show increased GSH concentration (*Baiz et al., 2014*). The intracellular decrease in GSH

concentration precedes the destruction of mitochondrial integrity, release of cytochrome c and caspase activation, and is recognized as an early step in apoptosis progression in response to various stimuli. Stimulation of glutathione synthesis can effectively protect cells against loss of mitochondrial membrane potential and inhibit apoptosis (*Zmorzyński et al., 2024*). *Baiz et al. (2014)* demonstrated that bortezomib induced a dose-and time-dependent increase in cell toxicity and decrease of cell viability with JHH6 (human HCC cell line JHH-6) being less sensitive than HepG2. *Hentze et al. (2002)* have shown that cancer cells maintain an increased GSH pool compared to that in normal tissue, and this has an impact on drug resistance. Cells containing a higher concentration of glutathione are also more resistant to apoptosis. Moreover, activation of caspase-apoptosis-inducing proteins requires an appropriate concentration of glutathione (*Robaczewska et al., 2016*). Our research shows that lowering the GSH concentration contributes to an increase in the induction of apoptosis in HepG2 cells. Oxidative stress affects intracellular antioxidants such as GPx (Fig. S2), GST (Fig. S3), GR (Fig. S4) well as CAT (Fig. S5), SOD (Fig. S6), and leads to MDA production (Fig. S7). *Cong (2019)* noted that changes in the concentration and activity of cellular antioxidants play a role in increasing susceptibility to bortezomib-induced apoptosis.

Glutathione peroxidase reduces phospholipid peroxides found in cell membranes and is critical in maintaining survival against oxidative stress (*Bułdak et al., 2014*; *Robaczewska et al., 2016*). Increased activity of GPx can aid in maintaining the net redox state within the malignant cells as a result of chemotherapy (*Salem et al., 2015*). In neuroblastoma cells, increased glutathione peroxidase activity promotes cytoprotection against proteasome inhibitors (*Michiels et al., 1994*). In the present investigation, recorded decrease in GPx, activity at two highest bortezomib concentrations (Fig. S2), which may prevent the drug resistance. Accordingly, bortezomib-resistant malignant cells show a higher activity of GPx (*Kalivendi et al., 2004*).

The GSTs enzymes are involved in protection of genome and cell organelles against the ROS (*Brigelius-Flohé & Maiorino, 2013*; *Allocati et al., 2018*). In the presented material we were able to show a decrease in glutathione S-transferase activity at all bortezomib doses (Fig. S3). Similar effect, but only at some bortezomib doses, concerns glutathione peroxidase. An important part of the defense system against oxidative damage constitute superoxide dismutase's. These enzymes catalyze superoxide anions dismutation and yield hydrogen peroxide and oxygen (*Allocati et al., 2018*). In this investigation we did not find correlation between MDA concentration and activities of the enzymes. Limitations of our study are associated with applied cell lines and methods. We found spontaneous apoptosis and necrosis in cell cultures without bortezomib (Figs. 1A and 1B), which may be due to the laboratory conditions including culture medium. The minimum essential medium Eagle could be used instead of William's medium. Another investigation could be carried out on additionally commercial hepatocellular carcinoma cell lines for example Hep3B or Huh7B12. Then obtained results could be compared between groups/cell lines. Unfortunately, these cell lines were not available for us. Another limitation of our study is the lack of ROS analysis. *Caillot et al. (2020)* have shown, that bortezomib at five nM and 10 nM increased the ROS levels in HepG2 cells. Given this data, we have focused our

analysis on other factors. Considering the limitations described above our results should be treated as the first step of the journey. However, it is important to note that we have performed many repetitions of the experiment to get reproducible data. Moreover, in the future we would like to expand the study including analysis of gene expression at mRNA level (with the use of real-time PCR) and protein level (with use of Western blot) to have a full picture of the changes in oxidative status of HepG2 cells.

## CONCLUSIONS

In our study, bortezomib affected the levels/activities of selected oxidative stress components depending on the dose of the applied drug and the duration of its action (24 h *vs.* 48 h). In general, the higher the dose of the drug and the longer the time of action, the more significant changes of studied parameters were observed.

## ACKNOWLEDGEMENTS

The authors thank Aleksandra Marcinkowska from the University of Zurich for preparing the graphical abstract and the reviewers for their useful comments.

### Funding

This work was supported by a program from the Minister of Education and Science called "Regional Initiative of Excellence" in the years 2019–2023, project no. 024/RID/2018/19. The funders had no role in study design, data collection and analysis, decision to publish, or preparation of the manuscript.

### Grant Disclosures

The following grant information was disclosed by the authors:
Minister of Education and Science called "Regional Initiative of Excellence": 024/RID/2018/19.

### Competing Interests

The authors declare there are no competing interests.

### Author Contributions

- Grażyna Świderska-Kołacz conceived and designed the experiments, authored or reviewed drafts of the article, and approved the final draft.
- Magdalena Madej performed the experiments, authored or reviewed drafts of the article, and approved the final draft.
- Szymon Zmorzynski conceived and designed the experiments, analyzed the data, prepared figures and/or tables, and approved the final draft.
- Wojciech Styk analyzed the data, prepared figures and/or tables, authored or reviewed drafts of the article, and approved the final draft.

- Iwona Surowiec performed the experiments, authored or reviewed drafts of the article, and approved the final draft.
- Bożena Witek performed the experiments, authored or reviewed drafts of the article, and approved the final draft.
- Anna Wojciechowska conceived and designed the experiments, prepared figures and/or tables, and approved the final draft.
- Joanna Czerwik-Marcinkowska conceived and designed the experiments, analyzed the data, authored or reviewed drafts of the article, and approved the final draft.
- Anna Nowakowska conceived and designed the experiments, analyzed the data, authored or reviewed drafts of the article, and approved the final draft.

## Data Availability

The raw data are available at Figshare: Czerwik Marcinkowska, Joanna (2025). Effects of Bortezomib - raw data. figshare. Journal contribution. https://doi.org/10.6084/m9.figshare.28550849.v1.

## Supplemental Information

Supplemental information for this article can be found online at http://dx.doi.org/10.7717/peerj.19235#supplemental-information.

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
