# Peer review of "Effects of bortezomib on intracellular antioxidant and apoptosis in HepG2cells"

_PeerJ, doi:10.7717/peerj.19235_

## Round 0.1 · original submission · Major Revisions

Although one of the reviewers recommended rejection, I decided to give you an opportunity to address the concerns of the reviewers and to amend manuscript accordingly.

·

Basic reporting

The paper is good and need correcting the revision that done

Experimental design

Well designed

Validity of the findings

They express results they obtained

Additional comments

See review report here
Review of the paper
"Effects of Bortezomib on Intracellular Antioxidant and Apoptosis in HepG2cells (#105961)
* * *
"Authors have chosen a topic; that express about work included
- Abstract: need more illustration and specification for determined parameters, in expression about results. You have to illustrate parameters that you speak about, like MDA and others.
*Also, the last two lines in abstract you said: Our study showed that the analyzed parameters changed after bortezomib treatment. ………..which analyzed parameters? Antioxidant or what?
* Also, Key words is too much, reduce it like
bortezomib; apoptosis; HepG2 cells; reduced glutathione; glutathione Sñtransferase,36 glutathione peroxidase; glutathione reductase; catalase; superoxide dismutase; malondialdehyde; oxidative stress
to
Bortezomib; Apoptosis; HepG2 cells; Oxidative stress.
With each beginning letter is capital
- Please, write all the main titles, like Introduction,………………Discussion, Bold, …..due to titles not clear, and wrote underneath titles bold?
- Under SOD, at lines142 , you wrote 0,3………… it is 0.3. and 145, you wrote 0,05 …….. it is 0.05.

- Under CAT, you have to add enzyme activity to all enzymes in tite and text. Also under CAT, line 152, need corrections, to the symbol of hydrogen peroxide, and many numbers attached to letters and words.
- Add GR to the title of GPx, due to you wrote its procedure, and not add it to the title.
- Under estimation of GSH: in calculation to mg protein? Please illustrate it ? and GSH, in non-protein thiol, so how you relate it to protein?
- Figure 1 legend………………not completed in writing
# I think that further studies are needed to overcome on multidrug resistance of bortezomib, like using another therapies or using anticarcinogenic and antioxidant promising natural product.
------------------------------------------------------------------------------------------------------------------------------------------thanks a lot

Reviewer 2 ·

Basic reporting

The manuscript presents the result of extensive work on the in vitro study of the effect of Bortezomib on the cell culture of hepatocellular carcinoma. The authors did not formulate the relevance of the study, but in principle it is understandable. However, unfortunately, I cannot recommend this manuscript for publication, since the conclusions do not correspond to the results, and the experimental work itself requires deep revision. Both the text and the presentation of the data require a complete revision. Below is a point by point explanation of my opinion:

Major points:
Abstract: there is no description of the relevance of the task, as well as it is not indicated what fundamental or applied significance the result has.

The hypothesis tested in the presented study is formulated by the authors as follows: «the activity of antioxidant enzymes and concentrations of reduced glutathione, as well as malondialdehyde depends on bortezomib doses and incubation time with this drug». At the same time, the authors of this work have previously shown that in the cells of patients with Multiple Myeloma, the activity of GSH and GPx is dose-dependent reduced by the action of Bortezomib (Figure 5. In https://doi.org/10.3390/genes15030387). Other studies also show that Bortezomib dose-dependently reduces intracellular glutathione levels in Multiple Myeloma cells (https://doi.org/10.1111/j.1365-2141.2008.07066.x). I see that this work performed in cells from another tumor, and I see an update on these and other data, but the authors should explain much more clearly why testing the formulated hypothesis is important for basic science and the development of approaches to the treatment of hepatocellular carcinoma.
Results. Lines 203-230: Presentation of cell death and MTT data are confusing. Were the flow cytometry analyses repeated at each time point and concentration? The proportions of events within each of the quadrants vary significantly from sample to sample in such an analysis. At least three times repetition (on individual cell suspensions) for each variant is necessary.
It seems to me that it is better to present the data in the form of charts, not tables. Primary dotplots It is better to move to Supplementary materials.
If the authors claim that "Flow cytometry analysis showed apoptosis induced by bortezomib in HepG2 cells in a time-dependent and dose-dependent manner" multivariate regression analysis should be carried out, not just the Chi square test.
I don't see the point in a direct comparison of the results of flow cytometry and MTT. The MTT results are better to presented as a separate chart with a transparent indication of the statistical power of the analysis and an ANOVA result with post hoc multiple comparison.

Results. Lines 231-242 and Figure 2. The assessment of morphology seems very superficial. The authors say that «The selected concentrations of bortezomib were minimum concentrations that did not cause changes in HepG2 cell morphology» and at the same time perform morphological evaluation. It is difficult to judge morphological changes from Figure 2, since the culture is photographed at very different levels of confluence. The authors state that «The severe morphological changes of cell death including rounding or shrinkage of cells, in a time dependent manner was not clearly observed (Figure 3)», but Figure 3 is about the level of reduced glutathione. No quantitative estimates of the proportion of living and dead cells can be made on the basis of single photographs of cultures per variant, without staining for viability, and even photographed at different confluence. This section of the results is not informative.

Results. Antioxidant system activity assessment: At what density (confluence) were the cells planted to assess the antioxidant system response to Bortezomib? I did not find this information in the materials and methods. Large differences between 24 and 48 hours in the control may be the result of oxidative stress occurring when the culture is overgrown.

Discussion. Lines 292-297. This is not a true statement. A correlation analysis between the parameters of antioxidant system activity and cell death has not been performed, but even if there is a correlation, this does not mean that «the factors are determinants of bortezomib cytotoxicity».

Discussion. Lines 332-333: «Our investigation confirmed that bortezomib is not toxic to hepatocytes at study concentrations.» - The statement does not correspond to the results of the study.

Conclusions.
Lines 397-398: No regression or correlation analysis were performed in the study.
Lines 400-401: In the introduction, another hypothesis was formulated. The hypothesis given here («the cells antioxidant enzymes play a key role in its response to bortezomib» ) has not been verified in any way in this study. The presence of changes in the antioxidant system does not mean that these changes play a key role in its response to bortezomib.
Lines 401-402: «It is possible that prolonged use of this drug would lead to the development of a multidrug resistance mechanism» - This part of the conclusion also does not follow from the results of the study.

Minor points:
Captions to the figures break off in the middle.
Abstract. Line 24-25. «All results were compared to the control groups included cells not treated with the drug». An obvious statement. There is no point in writing in the abstract.
Lines 75-77: The sentence is too general. Are we talking about tumor cells of patients or normal tissues? What specific tissues are we talking about? What does this have to do specifically with the experimental work presented in the manuscript?
Lines 83-84: Incorrect wording: «MDA concentration is a cytotoxic substance…»
A lot of missing spaces in the text.
Lines 129-137: Determination of cell viability (MTT test). It is necessary to indicate the starting number of cells per well of the plate and how it was estimated. MTT results are very much dependent on the starting number of cells (and how close they are to the plateau on the growth curve at the end of the analyzed period).
Line 208: I think, you mean: «The cells which were NOT stained with both dyes were alive (viable) and resided in region Q3».
Lines 208-211: It seems to me that it is better to transfer this to "Materials and Methods".
Line 234: I suppose it meant «24 h» but not «12 h».

Experimental design

The design of the study does not allow the conclusions stated in the manuscript to be drawn. More details in section 1 of the review.

Validity of the findings

The statistical analysis used does not allow the claims made in the study to be made. The cell death study does not have sufficient statistical power. More details in section 1 of the review.

---

## Round 0.2 · Minor Revisions

Please address the remaining concerns of the reviewer and amend manuscript accordingly

·

Basic reporting

Dear Editor

Authors do all corrections needed, I think that the paper could be published in your respectable Journal
I'm so happy for revising to your important Journal

Best Regards

Experimental design

Well designed

Validity of the findings

Good and accepted paper

Additional comments

To Authors

Many thanks for your effort for corrections

Best Regards

Reviewer 2 ·

Basic reporting

The fundamental comments (inconsistencies between data and conclusions) were taken into account by the authors. Corrections were made.

Minor points:
The caption to Figure 2 does not correspond to the figure. The MTT test results are discussed, and the cell fractions are presented according to the results of flow cytometry.

It is very strange that the authors transferred all the drawings of antioxidant enzyme activity to the Supplementary in the new version of the manuscript. In the first round of the review, I recommended moving the flow cytometry dotplots there, replacing them with diagrams of the average percentage of cells in different quadrants. But, the authors left the dotplots, but removed all activity graphs. Perhaps we just misunderstood each other. But this is more of an editorial issue. Not principled.

Experimental design

no comment

Validity of the findings

no comment

---

## Round 0.3 · accepted · Accept

All remaining issues were adequately addressed and the revised manuscript is acceptable now.